# Heat Shock Proteins and Breast Cancer

**DOI:** 10.3390/ijms25020876

**Published:** 2024-01-10

**Authors:** Miao Zhang, Xiaowen Bi

**Affiliations:** 1Department of Medical Genetics and Cell Biology, School of Basic Medical Sciences, Jiangxi Medical College, Nanchang University, Nanchang 330006, China; zhmiao19900310@163.com; 2Institute of Microbiology, Jiangxi Academy of Sciences, Nanchang 330096, China

**Keywords:** heat shock proteins, breast cancer, chaperone-assisted protein folding, post-translational modifications, heat shock protein inhibitors

## Abstract

Heat shock proteins (Hsps) are a group of stress-induced proteins involved in protein folding and maturation. Based on their molecular weight, Hsps can be divided into six families: small Hsps, Hsp40, Hsp60, Hsp70, Hsp90, and large Hsps. In the process of breast cancer tumorigenesis, Hsps play a central role in regulating cell reactions and functions including proliferation, metastasis, and apoptosis. Moreover, some of the critical Hsps also regulate the fine balance between the protective and destructive immunological responses within the tumor microenvironment. In this review, we systematically summarize the roles of major Hsps in breast cancer biology and point out the potential uses of these proteins in breast cancer diagnosis and therapy. Understanding the roles of different families of Hsps in breast cancer pathogenesis will help in the development of more effective prevention and treatment measures for breast cancer.

## 1. Introduction

Heat shock proteins (Hsps) are highly conserved, widely distributed, abundant proteins that are expressed in response to stress conditions, including carcinogenesis [1]. These proteins were first detected and identified in *Drosophila* tissues in 1962 [2], and later widely found in prokaryotic and eukaryotic cells. Hsps are a group of stress-induced proteins whose expression is strictly regulated by rapidly rising temperatures and which cannot be activated under normal physiological conditions. However, when organisms are subjected to a range of environmental and physiological insults, such as heat shock, oxidative stress, heavy metals, UV radiation, and membrane perturbation, heat shock proteins are rapidly activated and upregulated to aid cell survival or promote cell death due to irreparable damages [3]. Hsps have diverse biological functions, including monitoring and assisting in protein folding, preventing protein aggregation, facilitating the renaturation of denatured proteins, assembling multi-protein complexes, facilitating transmembrane transport and translocation, and participating in protein degradation processes [4].

Mammalian Hsps can be classified into six families based on their molecular weight: (1) small Hsps are encoded by the *HSPB* genes, and are less than or equal to 43 KD, such as Hsp27; (2) the Hsp40 family is encoded by *DNAJ* genes; (3) Hsp60 is encoded by *HSPD1* genes; (4) the Hsp70 family is encoded by *HSPA* genes; (5) the Hsp90 family is encoded by *HSPA* genes; and (6) the large Hsp family is encoded by *HSPH1* genes, including Hsp110 and GRP170 [5]. Apart from small molecule HSPs, Hsps are generally ATP-dependent proteins with ATPase activity. Heat shock factor (HSF) serves as an inducible transcriptional regulator of Hsps, and its phosphorylation is essential for the expression of the majority of Hsps. The heat shock element (HSE), which is a cis-acting sequence positioned upstream of the heat shock protein gene, can bind to and activate the expression of the heat shock protein gene [6]. Depending on the different molecular weights of the Hsps, the structures of different classes have divergent configurations. For example, Hsp60 uses barrel “Anfinsen” cage structures for the sequester folding of target proteins; Hsp70 and small Hsps have modular “clamps” to protect the extended hydrophobic structure in the target protein; and Hsp90 forms a multi-domain V-shaped structure whose scissor-like movement helps to refine the receptor protein [7]. In the process of tumorigenesis, Hsps play an important role in regulating cell reaction and function, and the expression of high levels of Hsps has been detected in a wide range of human tumors, including breast, endometrial, ovarian, gastric, colon, lung, and prostate cancers [8]. Indeed, some of the critical Hsps also intricately regulate the fine balance between the protective and destructive immunological responses within the tumor microenvironment [9].

In this review article, we focus on the relationship between breast cancer and the well-studied Hsp family members (Hsp27, Hsp40, Hsp60, Hsp70, Hsp90, and Hsp110). As shown in Table 1, the expression of Hsps (measured by FPKM value) is abnormally high in breast cancer cells, particularly Hsp27, Hsp70, and Hsp90, which suggests that high expression of Hsps is associated with breast cancer pathogenesis. Therefore, this article makes a systematic review of the important role of Hsps in breast cancer biology, which may provide innovative ideas for further research on breast cancer treatment.

## 2. The Biological Function of Hsps in Breast Cancer

### 2.1. Hsp27

Hsp27 (murine homolog Hsp25) is an ATP-independent protein that consists of an N-terminal WD/EPF domain, a conserved α-crystallin domain, and a flexible C-terminal domain [10]. Hsp27 has the ability to form large aggregates up to 800 kD; additionally, it can also create heteromeric structures with other Hsp family members, like Hsp20, and form complexes with additional proteins such as p38 mitogen-activated protein kinase (MAPK)-activated protein kinase 2 (MK2) and Akt. The process of oligomerization is regulated by phosphorylation. Following phosphorylation, Hsp27 undergoes oligomerization, predominantly forming dimers and tetramers, and establishes interactions with various proteins [11]. Hsp27 exists in the host body in two forms, phosphorylated and non-phosphorylated. Non-phosphorylated Hsp27 mainly recognizes and binds to proteins damaged by physical and chemical factors or misfolded, and cooperates with other Hsps to degrade host proteins through the proteasome pathway. On the other hand, phosphorylated Hsp27 is closely linked to host immune response, adapter protein activation, differentiation, proliferation and cell development. Studies have shown that Hsp27 is phosphorylated at the Ser15, Ser78, and Ser82 (corresponding to Ser86 in Hsp25) sites. There are numerous protein kinases shown to phosphorylate Hsp27 at these sites. MK2, MK3, PKB, and PKC contribute to the phosphorylation of Hsp27 at Ser15; MK2, MK3, MK5, PKC, and PKG phosphorylate Hsp27 at Ser78; MK2, MK3, MK5, PKB, PKD, and PKG cause the phosphorylation of Hsp27 at Ser82; PKG mediates Hsp27 indirectly, with activity towards Ser82 [12]. It has been verified that Hsp27 phosphorylation is associated with its translocation from the cytoplasm into the nucleus [13], and in addition, *O*-GlcNAcylation is required in the translocation of Hsp27 into the nucleus [14]. The structure of human Hsp27 is shown in Figure 1.

Hsp27 has been reported to be associated with the development of breast cancer. Hsp27 was found to be expressed at high levels in breast cancer tissues and MCF-7 lines, upregulating *HSPB8* by inducing the SUMOylation of HspB8, thereby facilitating the proliferation and metastasis of breast cancer cells [15]. Higher Hsp27 expression in the human breast could disrupt the Hippo tumor suppressor pathway, which has been reported to be related to proliferation restriction and apoptosis induction through an increase in the degradation rate of ubiquitinated mammalian STE20-like protein kinases 1 (MST1) [16]. In breast cancer stem cells (BCSCs), Hsp27 regulates the mammosphere-forming capability and cell migration potential through the activation of NF-κB [17]. Furthermore, in both breast cancer cells and antigen-presenting cells, elevated Hsp27 expression contributed to the immune escape mechanism of breast cancer and favored the differentiation of dendritic cells (DCs) to induce tolerance rather than response [18]. In addition, the phosphorylation of Hsp27 participates in the epidermal growth factor (EGF)-induced vasculogenic mimicry activity of BCSCs [19]. Ser78 of Hsp27 was phosphorylated in human epidermal growth factor receptor 2 (HER2)/neu-positive tumors, which could be an important effector in breast cancer development and progression [20]. Hsp27’s phosphorylation of Ser82 was critical for androgen receptor translocation from the cytoplasm to the nucleus and increased the proliferative ability of molecular apocrine breast cancer cells and the growth of xenograft tumors [21]. In addition, Hsp27 expression and its *O*-GlcNAcylation modification were associated with the malignant transformation of MCF-7 breast cancer cells [22]. In addition, there is compelling evidence suggesting that Hsp27 could be an attractive target for breast cancer therapy. In MDA-MB-231 breast cancer cells, Hsp27 conferred resistance to doxorubicin by decreasing or delaying the activation of apoptosis [23]. In the Herceptin-resistant SK-BR-3 breast cancer cells, Hsp27 reduced the susceptibility to Herceptin treatment by increasing HER2 protein stability [24]. In ERp29 over-expressing MDA-MB-231 cells and parental MCF-7 cells, Hsp27 has been supposed to be involved in ERp29-mediated resistance to doxorubicin [25]. The overexpression of Hsp27 in AS-B145 or BT-474 breast cancer cells diminished the inhibitory effect of ovatodiolide on mammosphere formation [26], while the phosphorylation of Hsp27 at Ser15 promoted its Ser78 phosphorylation and enhanced the nuclear localization of HER2 to induce Trastuzumab (TZMB) resistance in HER2^+^ breast cancer cells [27]. Hsp27-overexpressing KT breast cancer cells showed increased resistance to ultraviolet ray C (UVC) and interferon lethality [28], and our recent study also found that phosphorylated Hsp27 promoted ADR resistance in MCF-7 and MDA-MB-231 breast cancer cells by regulating DNA damage repair protein c-Myc dual phosphorylation [29]. A proposed model of Hsp27 in breast cancer development is illustrated in Figure 2.

As noted above, Hsp27 and its post-translational modifications (phosphorylation and GlcNAcylation) contribute to the occurrence and drug resistance of breast cancer. Therefore, Hsp27 inhibition in breast cancer appears as a promising breast cancer treatment. The depletion of Hsp27 stabilizes phosphatase and tensin homolog (PTEN), a tumor suppressor in MCF-7 breast cancer cells [30]. Hsp27 knockdown potentiates cleavage by caspase 3 and caspase 7, and can then sensitize MDA-MB-231 breast cancer cells to actinomycin D [31]. Methyl antcinate A exerts its inhibitory effect on BCSCs activity through the inhibition of Hsp27 expression [32]. Additionally, apatorsen (OGX-427), an antisense oligonucleotide (ASO) designed against Hsp27, has been assessed in phase II clinical trials (clinicaltrials.gov, accessed on 3 September 2023).

### 2.2. Hsp40

The Hsp40 (DNAJ) family is the largest and most diverse subgroup of the Hsp family and is classified into three DNAJ subclasses: DNAJA, DNAJB, and DNAJC [33]. DNAJA consists of an N-terminal J-domain, a glycine/phenylalanine (G/F)-rich region, a cysteine repeat (Cys-rich) region, and a largely uncharacterized C-terminus; DNAJB lacks the Cys-rich region and has an extended G/F rich region; and DNAJC differs substantially from DNAJA and DNAJB, as it lacks the G/F and Cys-rich region and the J-domain may be situated anywhere along the protein [34]. The structures of the DNAJ subclasses are shown in Figure 3A. The human genome encodes the expression of more than 41 different Hsp40 family proteins. Most of the Hsp40 family proteins contain the J domain, which binds to the interface between the nucleotide-binding and the substrate-binding domains of Hsp70; thus, Hsp40s are also known as Hsp70 co-chaperones [35]. Hsp90 plays a role in directing aggregation-prone proteins towards proper refolding through the Hsp70/Hsp40 system; in cases where the Hsp70/Hsp40 system fails, Hsp70 binds to Hsp90 to transfer the concomitant client, then the remodeled client is released from Hsp90 after spontaneous refolding [36,37]. An illustration of the current working model for the mechanism of chaperone-assisted protein folding by the Hsp70/Hsp40 complex in collaboration with Hsp90 is shown in Figure 3B.

Hsp40 family members have been found to have important roles in the progression of breast cancer. The expression of the *JDP1* (DNAJC12) gene was directly associated with the ER transactivation activity in breast tumors [38], and also upregulated in ER^+^ cultures of BCSCs [39]. The expressions of *DNAJA1* and *DNAJC9* genes were elevated in breast cancer samples and BCSCs, which is considered unfavorable for survival in breast cancer [40,41]. However, DNAJB4, whose overexpression resulted in cell apoptosis by activating the Hippo signaling pathway, deceased in triple-negative breast cancer (TNBC) patients and cells [42]. The protein level of DNAJB6 was decreased with the aggressiveness of breast cancer cells [43]. In addition, *DNAJC10* mRNA expression also reduced significantly in breast cancer cells (BT-20, MDA-MB-231, and ZR-75-1), and was associated with better overall survival and relapse-free survival in breast cancer [44].

Thus, it seems that the Hsp40 family may have dual functions in breast cancer, both as a tumor suppressor and as a tumor promoter. DNAJA3 negatively regulated the cell motility and metastasis of MDA-MB-231 breast cancer cells by inhibiting the FVIIa-induced transcriptional activity of IL-8 [45]. The depletion of DNAJA3 could inhibit P53 mitochondrial localization by absenting its direct interaction with P53, leading to resistance to the apoptosis of MCF-7 breast cancer cells under hypoxia or genotoxic stress [46]. It has been found that DNAJB6 negatively affected cancer properties, and a loss of DNAJB6 would lead to tumor growth, epithelial–mesenchymal transition (EMT), and metastasis by aberrantly activating Wnt/β-catenin signaling during breast cancer progression [47]. In contrast, some other Hsp40 family members have cancer-promoting activity; for instance, DNAJB1 suppresses P53-mediated apoptosis in A549 lung cancer cells [48], and DNAJB8 enhances the tumor-initiating ability of renal cell carcinoma [49]. However, their roles in breast cancer remain to be explored. Figure 4 presents some Hsp40 family members which are regarded as either tumor suppressors or tumor promoters. Given the roles of Hsp40 family members in breast cancer, it has also been demonstrated that they are involved in the pharmacodynamic effects of chemotherapy drugs. R115777, a farnesyltransferase inhibitor, suppressed the production of pro-angiogenic growth factors in BT-474 and MDA-MB-231 breast cancer cells by inhibiting the activities of DNAJA1 [50].

### 2.3. Hsp60

Hsp60 (HSPD) monomers are formed of three domains, including an apical domain, an intermediate domain, and an equatorial domain. Figure 5 shows a diagram depicting the structure of Hsp60 obtained from RCSB PDB. The apical domain binds to the co-chaperone and the substrate, which is implicated in ATP turnover; the intermediate domain acts as a connection hinge between the apical and the equatorial domain; and the equatorial domain encompasses the ATP-binding site, and facilitates the interactions between the single subunits within a ring and between the two heptameric rings of the chaperonin [51,52]. The subcellular localization of Hsp60 is crucial for understanding its functionality. While Hsp60 is commonly found in mitochondria, its accumulation has also been observed in the endoplasmic reticulum (ER) compartment during apoptosis [53]. Hsp60 was reported to have a variety of post-translational modifications, including phosphorylation, O-GlcNAcylation, N-Glycosylation, acetylation, and nitration, all of which, in turn, are very likely to endow Hsp60 with a wide range of functions [52].

It has been reported that the level of *Hsp60* expression in primary breast cancer is significantly higher compared with healthy breast tissues [54]. Hsp60 physically associates with P53 to block P53-dependent apoptosis, thereby regulating mitochondrial survival [55]. The ER-specific localization of Hsp60 plays a pro-apoptotic role by down-regulating anti-apoptotic protein XIAP expression in the chloroform fraction of *E. alba* (CFEA)-induced cell death [53]. In addition, Hsp60 has an important role in metastasization. Surface Hsp60 was associated with α3β1-integrin, a protein involved in the adhesion of metastatic breast cancer cells [56]. Hsp60 acts as an antigen of B and T lymphocytes in the early stage of breast cancer, and its autoimmunity was thought to be associated with an increased risk of transformation and tumor progression [9,54]. The exosomal Hsp60 level on LC3^+^ extracellular vesicles in the plasma of breast cancer patients is associated with disease progression and lung metastasis [57]. However, no significant correlation was found between the level of Hsp60 protein expression and tumor size or hormone receptor and HER-2 status in MDA-MB-435 breast cancer cells [58]. The role of Hsp60 in breast tumor apoptosis, metastasis, and immune modulation is illustrated in Figure 6.

In these latter instances, Hsp60 has a prognostic purpose and should be inhibited in breast cancer management. The interference of Hsp60 and HMGB1 could influence the growth of 4T1 breast cancer cells in an autocrine manner [59], while the inhibition of Hsp60 triggered cyclophilin D-dependent mitochondrial permeability transition and caspase-dependent cell death in MCF-7 breast cancer cells [60]. Mizoribine, an imidazole nucleoside, inhibited the chaperoning activity of the Hsp60–Hsp10 complex by interacting with Hsp60 to implement the immunosuppressive effect in breast cancer progression [51].

### 2.4. Hsp70

The Hsp70 (HSPA) superfamily consists of at least 13 members. Among them, the four major ones are constitutively expressed HSC70, ER-localized GRP70, mitochondrial mtHsp70, and stress-inducible Hsp70 [61]. The structure of Hsp70 consists of two parts, the N-terminal nucleotide-binding domain (NBD) and the C-terminal substrate binding domain (SBD), which are connected together through a flexible linker [62] (Figure 7). The NBD has a high affinity for ATP and ADP and consists of two lobes (I and II), which are further divided into four subdomains (IA, IIA, IB, IIB). There is a cleft between lobes I and II, and the nucleotide-binding site is located at the bottom of the cleft. The hydrolysis of ATP to ADP results in a conformational change in the NBD. The SBD can interact with hydrophobic amino acids in substrate molecules to bind substrate extension peptides. The SBD consists of a β-sandwich domain with a substrate binding site (β-SBD) and a flexible α-helical cap domain (α-SBD) capable of regulating the affinity of misfolded proteins. Hsp70 transiently associates with short hydrophobic peptide fragments within substrate proteins via the SBD, thereby assisting protein folding, and the cyclic process of substrate binding and release is driven by the ATP/ADP conversion [63].

Hsp70 is overexpressed in breast cancer patients and in various breast cancer cell line models, while the silencing of the gene encoding Hsp70 leads to inhibitory effects on cell proliferation, migration, invasion, and tumor growth of breast cancer cells [64]. The elevated production of Hsp70 in peripheral blood mononuclear cells may act as a diagnostic indicator for the prognosis of breast cancer patients [65]. The nuclear and cytoplasmic expression of Hsp70 correlated with better disease-free survival after ADR administration and showed a prognostic value in breast cancer patients [66]. In addition, circulating exosomal Hsp70 levels are inversely correlated with the response to the therapy, and monitoring changes in Hsp70 exosomes in the blood of cancer patients might be useful in predicting the breast tumor response [67]. The small extracellular vesicles (sEVs)-mediated Hsp70 transfer from MCF-7 breast cancer cells into the mitochondria of recipient cells conferred ADR resistance [68]. The IFNγ-induced Hsp70-exosome release from 4T1 breast cancer cells upregulated the costimulatory molecule expression of DCs, suggesting that tumor surveillance is dependent on the active release of Hsp70 from tumors [69]. It has been noticed that a strong protein expression of Hsp70 in almost all TNBC can even play an active role in inducing immunosuppression and tumor progression [70]. Hsp70, which is secreted from breast cancer cells, affects the pro-tumorigenic effects of tumor-associated macrophages, either directly or indirectly by inducing the expression of transforming growth factor (TGF)-β in breast cancer cells [71]. The elevated extracellular Hsp70 levels not only stimulated anti-tumor immune responses, but also induced immune tolerance in the case of chronic exposure [72]. It is generally held that elevated Hsp70 expression in breast cancer can function as a marker for poor prognosis. In addition, the post-translational modification of Hsp70 can also affect the development of breast cancer. Hsp70 acetylation disrupted the chaperone function of Hsp90, and then led to the downregulation of Hsp90 client proteins and induced apoptosis [73].

Given the breast cancer-promoting role of Hsp70, the inhibition of Hsp70 could bring benefits to breast cancer patients. The silencing of Hsp70 resulted in a tumor-specific death program of breast cancer cells (MDA-MB-468, MCF-7, BT-549, and SK-BR-3), but did not affect non-tumorigenic breast epithelial cells or normal human fibroblasts. The tumor-specific death program of breast cancer cells was independent of caspases and bypassed Bcl-2 [74]. To date, some small molecule inhibitors and peptide inhibitors of Hsp70 have been verified with hopeful effects in breast cancer models. For instance, the inhibition of Hsp70 expression by epigallocatechin-3-gallate decreased cell proliferation and colony formation in MCF-7 breast cancer cells [75]. Sulphoraphane, a natural isothiocyanate inhibitor of Hsp27, Hsp70, and Hsp90 expression, induced apoptosis in MCF-7 and MDA-MB-231 breast cancer cells [76]. Monobenzyltin complex C1 decreased the expression of Hsp70, thereby inducing cell apoptosis in MCF-7 cells [77]. Crocin has anti-proliferative and apoptotic effects on MDA-MB-468 breast cancer cells by reducing the expression of Hsp70 [78]. Gold nanoparticles reduced the expression of Hsp70 in the nucleoli to enhance the killing of MCF-7 breast cancer cells [79]. In addition, by downregulating Hsp70 expression, azacytidine increased the sensitivity of MCF-7 breast cancer cells to adriamycin [80]. Methylene blue, an Hsp70 ATPase inhibitor, reduced cell migration by inhibiting matrix metalloproteinase (MMP)-2 activation in MDA-MB-231 breast cancer cells [81]. Inhibition of Hsp70 ATPase by VER-155008 in MCF-7 breast cancer cells effectively induced apoptosis in a time-dependent manner [82]. The inhibiting Hsp70 ATPase through piperidine derivatives or thiourea derivatives showed potent anti-cancer activities against lapatinib-resistant breast cancer cells [83,84]. When breast cancer cells were challenged with MAL3-101 to reduce Hsp70 activity, the unfolded protein response and apoptosis were increased in breast cancer cells [85]. Reducing the affinity of Hsp70 to ATP using HS-72 reduced tumor growth in a spontaneous mouse mammary tumor model [86]. Blocking the interaction between HSC70 and BAG-1 via BAG-1-derived peptides inhibited cell proliferation in MCF-7 and ZR-75-1 breast cancer cells [87]. When the interaction between Hsp70 and BAG-3 was broken by benzothiazole-rhodacyanines JG-98 or JC-231, apoptosis and necroptosis were increased in MDA-MB-231 breast cancer cells and a xenograft model [88,89]. Figure 8 illustrates the effects of the above mentioned Hsp70 inhibitors on breast cancer. However, so far none of these inhibitors on the expression or activities of Hsp70 have either been approved for clinical oncology or demonstrated encouraging results in clinical trials due to the fact that they also affect the viability of normal cells.

### 2.5. Hsp90

Hsp90 (HSPC) consists of three domains: an N-terminal domain, a middle domain, and a C-terminal domain (Figure 9). In the absence of ATP binding, Hsp90 exists mainly in the “V”-type open conformation. The N-terminal domain is a dimeric structure and contains the ATP-binding site [90]. Hsp90, together with its accessory molecular chaperone, regulates the hydrolysis of ATP and prompts the initiation of the Hsp90 molecular chaperone cycle to provide energy for the process [91]. The middle domain is the binding region of the substrate protein and the accessory molecular chaperone. The N-terminal domain and the middle region are connected by a segment of a conformationally variable and charged linker, which brings great difficulties to the specific resolution of the full-length structure of Hsp90 due to its structural instability. Hsp90 possesses ATPase activity only when the catalytic loop in the middle domain becomes open and activated. The loop structure in the middle domain contains a conserved Arg residue that interacts with the γ-phosphate of ATP to facilitate Hsp90-regulated ATP hydrolysis for energy supply. The C-terminal domain is another accessory chaperone binding region and is responsible for the dimerization of Hsp90 [37]. The C-terminal domain has a conserved MEEVD sequence that interacts with a tetratricopeptide repeat (TPR)-containing an accessory chaperone [92]. After the substrate protein binds to the middle domain, Hsp90 regulates the conformational rearrangement of the N-terminal domain through interactions with the accessory molecular chaperone and the water release of ATP, and, finally, the conformation of Hsp90 changes to the “off” state, at which time Hsp90 can function as a molecular chaperone [93]. Moreover, the chaperone activity of Hsp90 was known to be regulated by several post-translational modifications, including phosphorylation, acetylation, oxidation, and S-nitrosylation [94].

Hsp90 has been demonstrated to regulate the stability of client proteins, such as HER2, CD4, AKT, RAF-1, and Bcr-Abl, most of which are essential for cancer cell proliferation. Notably, HER2 is a well-defined client protein in breast cancer, and Hsp90 is expressed at higher levels in breast cancer cells than in normal cells. High Hsp90 expression in primary breast cancer has shown a strong association with decreased survival [95]. In multiple breast cancer datasets, the high expression of Hsp90α was significantly associated with poor prognosis, including a high histological grade, higher Nottingham prognostic index (NPI) scores, and a loss of hormone receptor expression [96]. With inverse correlation to the total expression level, the glutathionylated status of Hsp90 can subsequently enhance its degradation to impede the binding of Hsp90 with its client proteins, thereby correlating with favorable prognosis in breast cancer treatment [97]. The C-terminal domain of Hsp90 interacted with phosphoglycerate kinase 1 (PGK1), thereby facilitating the binding of GSK3β and Hsp90 to stabilize GSK3β expression, which is critical for the stemness maintenance of BCSCs [98]. In MCF-7 and MDA-MB-231 breast cancer cells, Hsp90 was involved in the stabilization and nuclear accumulation of E2F1 and E2F2, which may govern tumor progression [99]. In addition, a comprehensive bioinformatics analysis of breast cancer exosomes revealed Hsp90AA1 as a novel hub gene in the exosomal proteins derived from human breast cancer cells [100].

It has been reported that triple-negative breast cancer (TNBC) is sensitive to the Hsp90 inhibitor PU-H71 through the inhibition of the RAS/RAF/MAPK pathway and the induction of apoptosis via the degradation of AKT and BCL-XL [101]. Moreover, another Hsp90 inhibitor, DCZ3112, could exert anti-tumor activity against HER2-positive breast cancer by disrupting the Hsp90-Cdc37 complex [102]. The inhibition of Hsp90 could decrease the YAP protein in TNBCs and is a promising strategy for treating TNBC breast cancers [103]. In fact, Hsp90 inhibitors targeting different domains of Hsp90 exhibit distinct regulatory mechanisms (Figure 10). The N-terminal domain Hsp90 inhibitors could promote the lysosomal degradation of oncogenic ATPase MORC2 through the disruption of its homodimer formation, thereby suppressing MORC2-driven breast cancer progression [104]; in addition, disrupting the interaction between Hsp90 and PGK1, and reducing GSK3β expression, resulted in a significantly reduced inhibition of β-catenin expression to maintain the stemness of breast cancer stem cells [98]. However, it has been proposed that N-terminal domain Hsp90 inhibitors would trigger the translocation of the transcription factor HSF-1 to the nucleus, leading to further upregulation of Hsp70, Hsp40, and other Hsps, followed by the activation of the survival cascade known as the heat shock response. Therefore, none of the N-terminal domain Hsp90 inhibitors have succeeded to date [105]. A novel C-terminal domain Hsp90 inhibitor, SL-145, effectively suppressed metastatic triple-negative breast cancers by inhibiting oncogenic AKT, MEK/ERK, and JAK2/STAT3 signaling, and without an induction of the heat shock response [106].

It should be apparent that Hsp90 is an exciting new therapeutic target, and that its inhibition delivers a combinatorial attack on multiple oncogenic pathways and many hallmark traits of breast cancer. Even so, no Hsp90 inhibitors have been conducted in phase III trials and no approval has been given for their use in clinical practice.

### 2.6. Hsp110

Hsp110 (also known as HSP105 or HSPH1), a member of the Hsp70 superfamily, mainly consists of an N-terminal ATPase domain with similarity to actin, a central β-sheet domain directly binding peptide substrates, and a C-terminal α-helix domain homologous to those of the Hsp70 family regulating its substrate binding (Figure 11A) [107]. It has been reported that Hsp110 acts as a diverged subgroup of the Hsp70 family; for instance, Hsp110 can interact with Hsp25 and Hsp70 [108], and Hsp110 synergizes with Hsp70 and Hsp40 in mammalian cytosol to maintain cellular homeostasis [109]. Hsp110–Hsp70–Hsp40 cooperation in protein folding forms a reaction cycle. Firstly, unfolded proteins are recognized and recruited to Hsp70 with the assistance of Hsp40, and Hsp70 undergoes a dramatic structural rearrangement, which leads to tight substrate binding. Subsequently, Hsp110 binds to Hsp70 and induces ADP release from Hsp70; meanwhile, Hsp110 binds directly with unfolded substrate proteins to further prompt the recognition of the unfolded substrates by Hsp110. Finally, the Hsp70∙ATP-substrate complex is dissociated to allow partial or complete folding after ATP binding to Hsp70, while the partially folded substrate rebinds to Hsp70 for a new cycle (Figure 11B) [110]. Hsp110 can also be phosphorylated and exerts specific biological effects. It has been reported that protein kinase CK2 phosphorylated Hsp105α at Ser509 in the β-sheet domain and abolished the inhibitory activity of Hsp105α on the Hsc70 chaperone activity [111].

Hsp110 was overexpressed in breast cancer, which could serve as a prognostic biomarker correlating with poor prognosis for patients with breast cancer. When Hsp110 was depleted, the integration of protein phosphatase 2A (PP2A) was disrupted into β-catenin degradation, compromising cancer cell proliferation [112]. It is interesting to note that Hsp110 acted as a “danger signal” by interacting with mouse mammary carcinomas to induce anti-tumor immune responses [113]. However, the rate at which the tumor proliferates is actually higher than the rate at which it is killed by the immune response [114]. As a result, the immune response may not be sufficient to eliminate tumors. However, a complex formed by Hsp110 with the intracellular domain (ICD) of human HER-2/*neu* could be used to prepare vaccines against spontaneous mammary tumors in FVB-*neu* (FVBN202) transgenic mice [115].

Altogether, owing to the high chaperoning and immunological activity, recombinant Hsp110 is very promising for breast cancer immunotherapy.

## 3. Conclusions

This review has attempted to describe the role of the well-studied Hsp family members including Hsp27, Hsp40, Hsp60, Hsp70, Hsp90, and Hsp110 in breast cancer. As summarized in Table 1, the expression of most Hsps, in particular Hsp27, Hsp70, and Hsp90, are abnormally high in breast cancer cells. Although Hsps are classified based on their molecular weights, most Hsps have similar functions in carcinogenesis, the prevention of apoptosis, and in conferring drug resistance. Among Hsps proteins, Hsp60, Hsp70, and Hsp90 are present on the surface of tumor cells and are released by the cells through exosomes, making them potential prognostic biomarkers that are easily accessible. Similarly, Hsp27, Hsp40, and Hsp110 are also associated with breast cancer development. However, Hsp40 has been reported to have dual functions in breast cancer; in addition, Hsp110 is mainly considered an adjuvant to breast cancer antigens in immunotherapy.

Therefore, the pharmacological inhibition of Hsp27, Hsp60, Hsp70, and Hsp90 may provide therapeutic opportunities for breast cancer treatment. However, no Hsps inhibitors have yet been approved by the FDA for clinical use in patients with breast cancer. The efficacy of current Hsps inhibitors is relatively low, and the inhibition of one Hsps member could upregulate other Hsps via negative feedback to weaken the overall therapeutic effect of the inhibitors. For instance, the inhibition of the N-terminal domain of Hsp90 stimulates the heat shock response, resulting in further upregulation of Hsp70, Hsp90, and other Hsps. As a result, no N-terminal domain Hsp90 inhibitors have succeeded to date. Therefore, targeting Hsp27, Hsp60, Hsp70, or Hsp90 with CRISPR/Cas9 to enhance the selectivity for target Hsps, and combining different Hsps inhibitors, could be promising strategies to improve the effectiveness of cancer therapy. Hsps are also required in normal cells to maintain cellular homeostasis. However, the difference in the amounts of Hsps in normal cells and breast cancer cells is still not fully understood. In response to this problem, we can improve the accuracy of drug delivery to breast cancer-specific tissue locations by using nanoparticles carrying Hsps inhibitors.

In summary, understanding the functions and molecular mechanisms of Hsps is crucial for improving the accuracy of breast cancer diagnosis and developing more effective chemotherapeutic agents. Additionally, using some Hsps as adjuvants to breast cancer antigens may show promising results and should be further investigated in the near future.

## Figures and Tables

**Figure 1 ijms-25-00876-f001:**
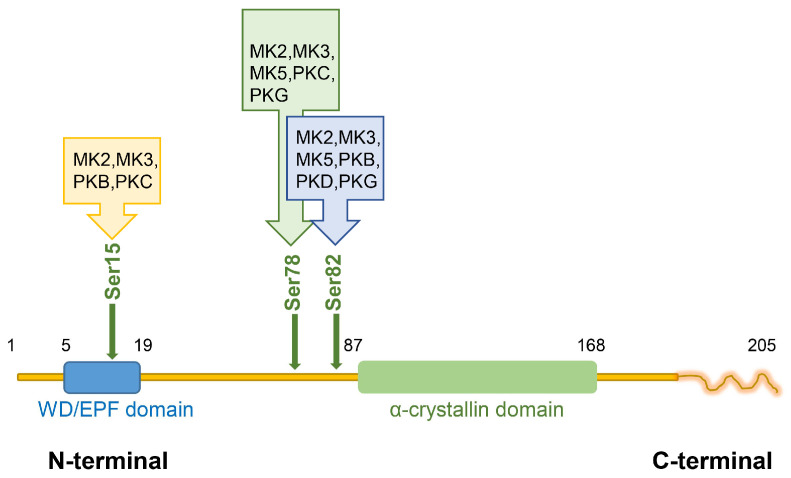
The structure of human Hsp27. The structure of human Hsp27 includes an N-terminal domain, an α-crystallin domain, and a C-terminal domain. The phosphorylated serine residues at Ser15, Ser78, and Ser82 are also shown. Ser15 can be phosphorylated by MK2, MK3, PKB, and PKC; Ser78 by MK2, MK3, MK5, PKC, and PKG; and Ser82 by MK2, MK3, MK5, PKB, PKD, and PKG.

**Figure 2 ijms-25-00876-f002:**
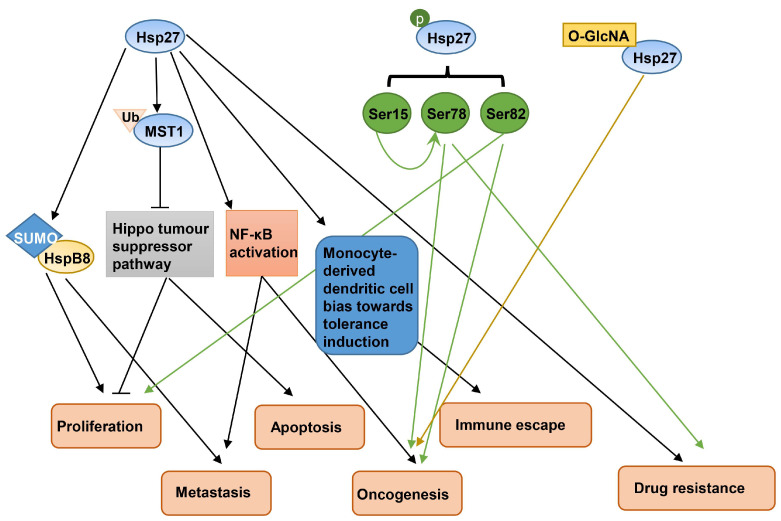
Different post-translational modifications of Hsp27 in the development of breast cancer. Hsp27 and its post-translational modifications (phosphorylation and GlcNAcylation) are associated with the proliferation, metastasis, apoptosis, oncogenesis, immune escape, and drug resistance of breast cancer.

**Figure 3 ijms-25-00876-f003:**
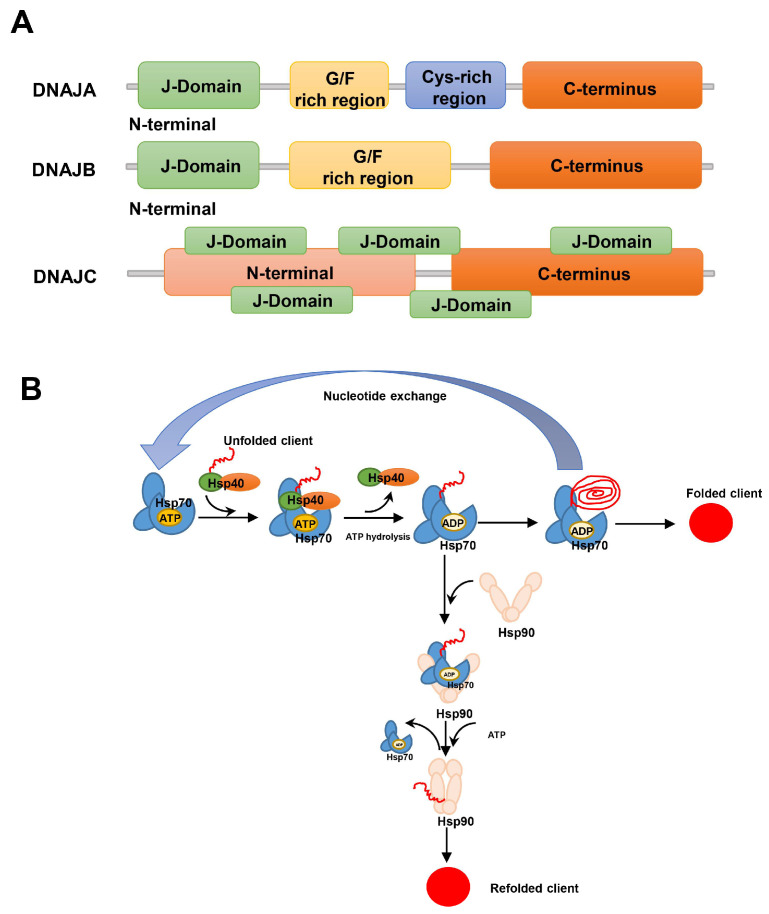
Basic structure and working model for the Hsp40 family. (**A**) The Hsp40 family was classified into three subclasses according to the presence or absence of three domains, including the J domain, a G/F rich region, and the Cys-repeats motif, together with a C-terminal domain that is largely uncharacterized. (**B**) Working model for the mechanisms of chaperone-assisted protein folding by the Hsp70/Hsp40 complex in collaboration with Hsp90. The unfolded client interacts with Hsp40, which then binds to the ATP-bound form of Hsp70 to deliver the unfolded client to the substrate-binding cleft of Hsp70. Binding with Hsp40 stimulates the hydrolysis of ATP to ADP by the ATPase in Hsp70. The unfolded protein is prevented from aggregation and/or undergoes partial remodeling through repeating cycles of client binding and regeneration of the ATP-bound form of Hsp70. Alternatively, the ADP-bound form of Hsp70 interacts with Hsp90 and then promotes client transfer from Hsp70 to Hsp90. At the end, Hsp90 releases the refolded client.

**Figure 4 ijms-25-00876-f004:**
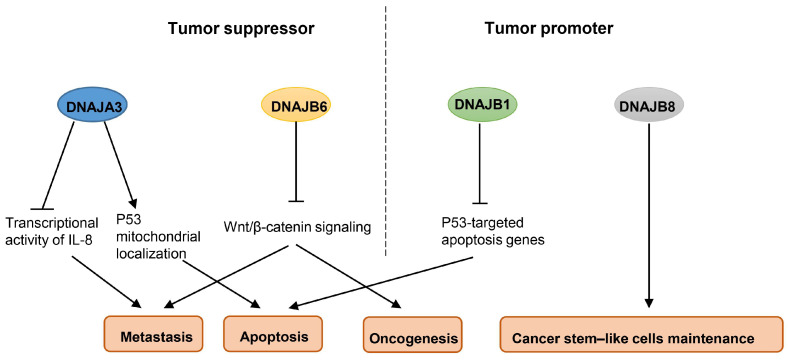
The functions of the Hsp40 family in breast cancer cell regulation. DNAJA3 and DNAJB6 exert a tumor suppressor effect; DNAJB1 and DNAJB8 exert a tumor promoter effect.

**Figure 5 ijms-25-00876-f005:**
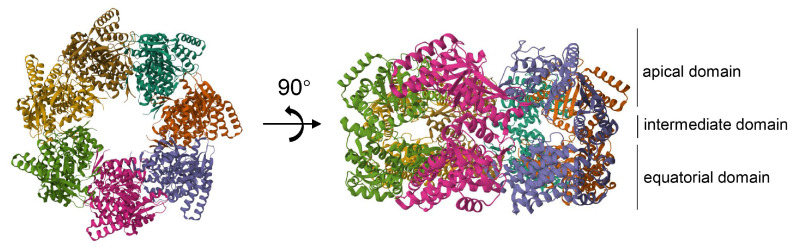
Three-dimension structure of Hsp60. Heptamer monocyclic structure of Hsp60 (PDB:7AZP, obtained from RCSB PDB). Hsp60 monomer has three structural domains: apical domain, intermediate domain, and equatorial domain. The apical domain binds to the co-chaperone and the substrate, the equatorial domain encompasses the ATP-binding site, and the intermediate domain acts as a connection hinge between the apical and the equatorial domain.

**Figure 6 ijms-25-00876-f006:**
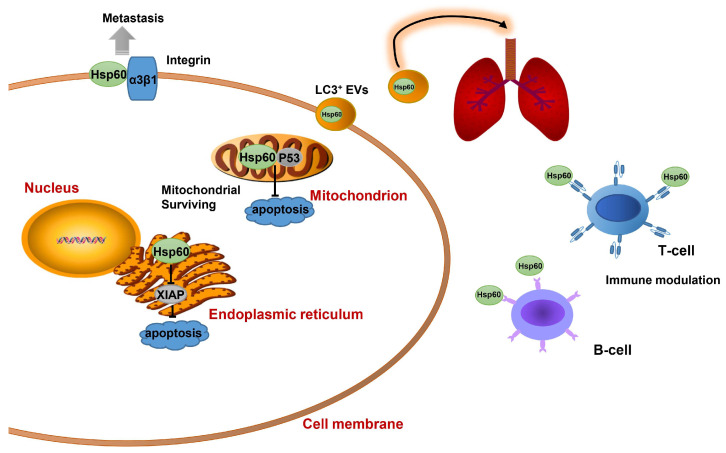
A model illustrating Hsp60’s regulation of breast cancer. Hsp60 in the endoplasmic reticulum inhibits XIAP expression to confer resistance to apoptosis. Mitochondrial Hsp60 interacts with P53 to suppress P53-dependent apoptosis. Hsp60 on the cell membrane interacts with α3β1-integrin to promote metastasis. Extracellular Hsp60 acts as an antigen for B-cells and T-cells to modulate the immune system in early stages of breast cancer. Exosomal Hsp60 in plasma accelerates the lung metastasis of breast cancer.

**Figure 7 ijms-25-00876-f007:**
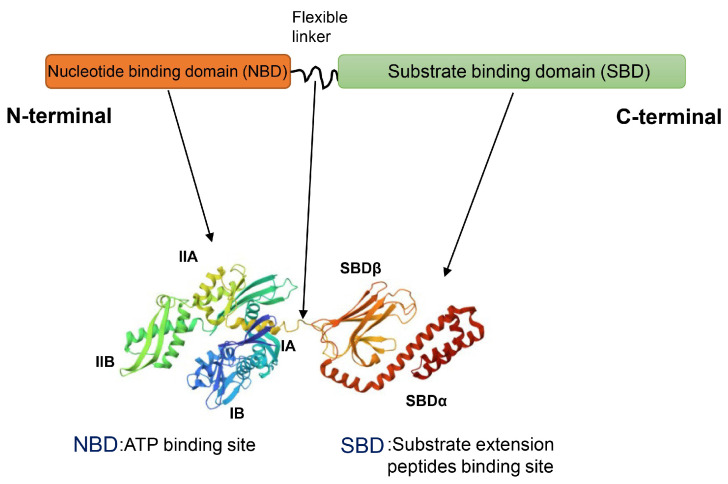
The structure of Hsp70. The Hsp70 structure consists of two conserved functional domains, the N-terminal nucleotide-binding domain (NBD) and the C-terminal substrate-binding domain (SBD), which are connected via a flexible linker. NMR-RDC/XRAY structure of *E. coli* Hsp70 (DNAK) chaperone (1-605) complexed with ADP and substrate (PDB:2KHO, obtained from RCSB PDB). The NBD consists of four subdomains (IA: dark green, IIA: yellow, IB: blue, IIB: fluorescent green) and, as the ATP binding site, exhibits ATPase activity. The SBD consists of β-SBD (orange) and α-SBD (red) and is the substrate extension peptides binding site.

**Figure 8 ijms-25-00876-f008:**
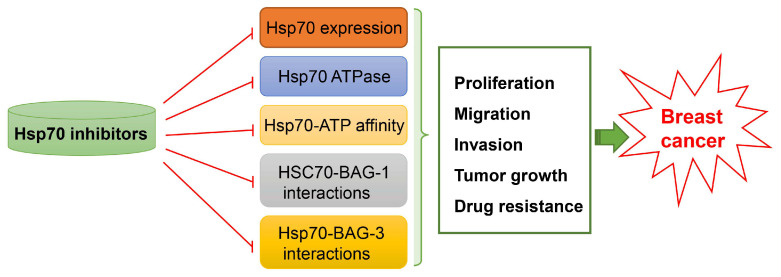
Simplified scheme showing the therapeutic effects on breast cancer exerted by inhibitors targeting Hsp70. Hsp70 inhibitors targeting Hsp70 expression, Hsp70ATPase, Hsp70-ATP affinity, HSC70-BAG-1 interactions, or Hsp70-BAG-3 interactions have promising effects on breast cancer by regulating its proliferation, migration, invasion, tumor growth, and drug resistance.

**Figure 9 ijms-25-00876-f009:**
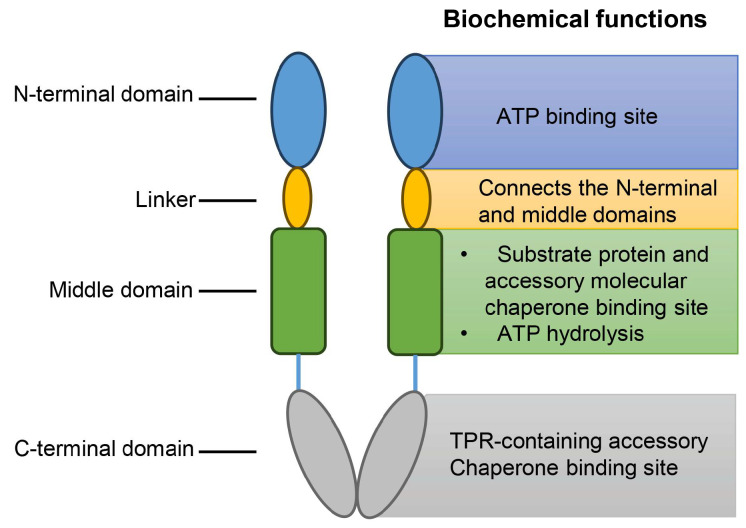
The basic structure of Hsp90 dimer. The Hsp90 structure consists of three functional domains: the N-terminal domain, the middle domain, and the C-terminal domain. Before ATP binding, Hsp90 exists mainly in the “V” type open conformation. The N-terminal domain and the middle domain are connected by a flexible linker. The N-terminal domain is the binding site for ATP; the middle domain interacts with the substrate protein, the accessory molecular chaperone, and the γ-phosphate of ATP to facilitate the Hsp90-regulated ATP hydrolysis for energy supply; and the C-terminal domain is the base for Hsp90 dimerization, as well as the site for TPR-containing accessory chaperone binding.

**Figure 10 ijms-25-00876-f010:**
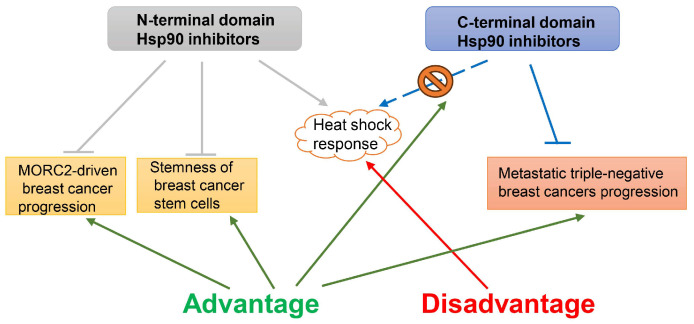
Simplified scheme showing the advantages and disadvantages of the existing N-terminal domain and C-terminal domain Hsp90 inhibitors as breast cancer therapeutics. The N-terminal domain Hsp90 inhibitors show advantages in breast cancer therapy by suppressing MORC2-driven breast cancer progression and inhibiting the stemness of breast cancer stem cells, but have the disadvantage of triggering heat shock response, while the N-terminal domain Hsp90 inhibitors could suppress metastatic triple-negative breast cancers without causing heat shock response.

**Figure 11 ijms-25-00876-f011:**
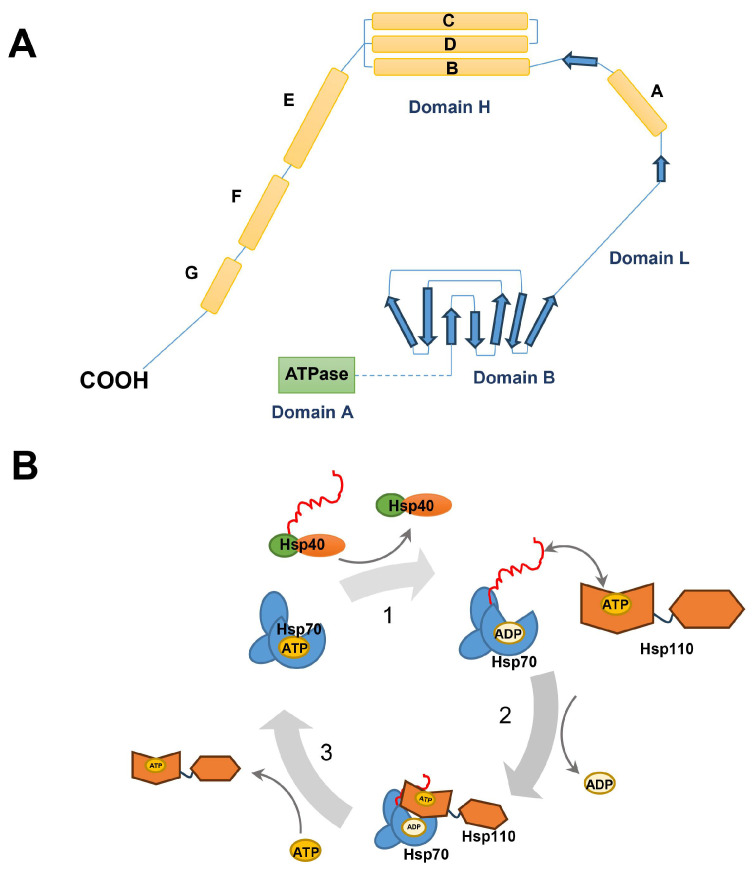
Proposed folding pattern and working model for Hsp110. (**A**) The structure of Hsp110 includes A, B, L, and H domains. Domain A is the N-terminal ATPase domain, domain B is the peptide-binding domain, domain H is the C-terminal domain including the helix domain, and domain L is the linker between domain B and domain H. The blue arrows represent β-strands. (**B**) Model for Hsp110–Hsp70–Hsp40 cooperation cycle. (1) Recruitment of Hsp70 to unfolded peptide substrates is assisted by Hsp40. (2) Complex formation between Hsp70 and Hsp110 displaces ADP from the Hsp70 partner. (3) Hsp70–Hsp110 complex dissociates, and the substrate protein is released for folding after ATP binding to Hsp70.

**Table 1 ijms-25-00876-t001:** Differential Hsps expression in normal and BRCA tissue. The data were retrieved from The Cancer Genome Atlas (TCGA) using the Gendoma web server (https://ai.citexs.com, accessed on 11 August 2023). FPKM, Fragments Per Kilobase of transcript per million mapped reads, represents the number of fragments matching the specific gene’s 1 kb-long exon region per million sequencing fragments, and is used to express the relative expression level of genes. At a FPKM ≥ 200, 6 genes were selected and defined as highly-expressed genes, including *HSPB1*, *HSPA5*, *HSPA8*, *HSP90AA1*, *HSP90AB1*, and *HSP90B1*.

Hsp Family	Gene	Normal Type	BRCA Type	Hsp Family	Gene	Normal Type	BRCA Type
		(FPKM)	(FPKM)			(FPKM)	(FPKM)
Hsp27	*HSPB1*	117.625	352.549	Hsp40	*DNAJC17*	2.77648	3.03727
	*HSPB2*	4.17935	0.795792		*DNAJC18*	2.74528	1.37358
Hsp40	*DNAJA1*	60.6343	89.2385		*DNAJC19*	7.09079	10.2867
	*DNAJA2*	17.6651	16.4556		*DNAJC20*	7.39097	7.49364
	*DNAJA3*	6.61321	11.5337		*DNAJC21*	6.50134	7.68681
	*DNAJA4*	6.20316	12.74		*DNAJC22*	1.48905	3.92159
	*DNAJB1*	51.4863	49.5621		*DNAJC23*	17.9754	18.5166
	*DNAJB2*	14.7851	18.5249		*DNAJC24*	1.57841	1.38519
	*DNAJB3*	0.00184269	0.00143597		*DNAJC25*	4.83016	4.64115
	*DNAJB4*	13.8392	5.21584		*DNAJC26*	4.61191	6.98525
	*DNAJB5*	1.98587	1.57866		*DNAJC27*	2.26955	1.39313
	*DNAJB6*	7.20102	8.04553		*DNAJC28*	0.858132	0.706425
	*DNAJB7*	0.0585917	0.056792		*DNAJC29*	3.08858	1.35963
	*DNAJB8*	0.00133167	0.0017791		*DNAJC30*	5.94151	6.78276
	*DNAJB9*	14.9374	14.5789	Hsp60	*HSPD1*	60.7563	93.6693
	*DNAJB11*	7.70387	13.5074	Hsp70	*HSPA1A*	36.4994	54.2845
	*DNAJB12*	11.4622	12.1393		*HSPA1B*	39.2733	48.0456
	*DNAJB13*	0.0583365	0.116631		*HSPA1L*	1.29012	1.49757
	*DNAJB14*	3.77468	3.37966		*HSPA2*	5.03015	9.5758
	*DNAJC1*	13.7391	39.0695		*HSPA4*	25.4501	38.5318
	*DNAJC2*	4.11542	5.8259		*HSPA4L*	1.85338	1.29703
	*DNAJC3*	20.8626	21.6129		*HSPA5*	123.308	210.117
	*DNAJC4*	10.4769	12.6855		*HSPA6*	1.27087	3.92941
	*DNAJC5*	16.7853	23.3509		*HSPA7*	2.43346	3.12447
	*DNAJC5B*	0.186127	0.430458		*HSPA8*	189.73	263.496
	*DNAJC5G*	0.0200045	0.0112469		*HSPA9*	53.5377	74.6974
	*DNAJC6*	0.5795	0.945219		*HSPA12A*	6.20892	1.81148
	*DNAJC7*	6.90192	8.76527		*HSPA12B*	8.11172	3.32468
	*DNAJC8*	32.1043	30.8205		*HSPA13*	8.29694	11.8282
	*DNAJC9*	3.19184	5.79583		*HSPA14*	4.70054	7.35153
	*DNAJC10*	5.90804	7.9543	Hsp90	*HSP90AA1*	222.658	304.964
	*DNAJC11*	6.00964	6.45572		*HSP90AB1*	416.171	611.813
	*DNAJC12*	5.79399	23.0184		*HSP90B1*	151.788	202.219
	*DNAJC13*	8.89732	8.54604		*HSP90B2P*	0.120954	0.119044
	*DNAJC14*	5.97507	7.31624		*HSP90L*	10.2401	14.1013
	*DNAJC15*	5.41503	5.17456	Hsp110	HSPH1	12.0038	21.2046
	*DNAJC16*	4.415	4.81457				

## Data Availability

Data are contained within the article.

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
