# Peer review of "Heat Shock Proteins and Breast Cancer"

_ijms, 2024, doi:10.3390/ijms25020876_

Round 1

Reviewer 1 Report

Comments and Suggestions for Authors

The manuscript " Heat Shock Proteins and Breast Cancer" by Zhang M and Bi X, aims to describe the current knowledge on heat shock proteins in cancer, particularly in breast tumours, both at a diagnostic/prognostic and therapeutic level.

The review appears to be a generous attempt to provide a comprehensive overview of the topic. However, the paragraphs describing the function of each family member have given rise to a chaotic annotation of what has been documented in the literature (most of the time relating to observations made using in vitro rather than in vivo models), with little critical revision that motivates the role of these proteins only on the base of syllogistic reasoning. This makes it extremely difficult (if not impossible) for the reader to understand what the original contribution of the manuscript could be in cultural terms.

In light of the above, the reviewer believes that this manuscript may have a very limited cultural impact as it adds nothing to what has already been reported by other authors in similar reviews (see: Alberti G, published in July 2022 by this journal). This drastically reduces interest in the manuscript's eligibility for publication in Int. J Molecular Science.

Comments on the Quality of English Language

English grammar is poor and deserves a major overhaul.

Author Response

Responses to Reviewer #1:

Comment 1: The manuscript " Heat Shock Proteins and Breast Cancer" by Zhang M and Bi X, aims to describe the current knowledge on heat shock proteins in cancer, particularly in breast tumours, both at a diagnostic/prognostic and therapeutic level.

The review appears to be a generous attempt to provide a comprehensive overview of the topic. However, the paragraphs describing the function of each family member have given rise to a chaotic annotation of what has been documented in the literature (most of the time relating to observations made using in vitro rather than in vivo models), with little critical revision that motivates the role of these proteins only on the base of syllogistic reasoning. This makes it extremely difficult (if not impossible) for the reader to understand what the original contribution of the manuscript could be in cultural terms.

In light of the above, the reviewer believes that this manuscript may have a very limited cultural impact as it adds nothing to what has already been reported by other authors in similar reviews (see: Alberti G, published in July 2022 by this journal). This drastically reduces interest in the manuscript's eligibility for publication in Int. J Molecular Science.

Response: We thank the reviewer for providing a timely review of our manuscript.

In the Section 2 of this review, we first described the structure of each member of heat shock proteins (Hsps), then described their post-translational modifications, and the association between Hsps and breast cancer development (the order of description takes into account the subcellular localization where Hsps exerts their function in breast cancer regulation, the post-translational modification of Hsps in regulating breast cancer development , the order of publication of the references, etc.), and finally, we proposed the role of the Hsps in breast cancer and described the application of existing interference or inhibitors of the Hsps.

In fact, some of the references cited in this review have carried out in vivo mechanism verification in mice, but we believe that the pathogenesis of the disease is ultimately explained by the role of intracellular proteins. Therefore, in this review, we anchored our discussion mainly to the cellular level and summarize the role of heat shock proteins in breast cancer cells. We used relatively few critical languages in the Section 2 in order to truthfully overview the findings of the original research cited. However, most of the critical language was found in Section 3(Conclusion), where we not only systematically summarized the relationship between heat shock protein and breast cancer, but also pointed out the problems existing in the prevention and treatment of breast cancer through targeting Hsps, and finally we put forward a prospect for the field.

Although a review published by Alberti G et al in 2022 was on the same topic, our article differs from that of Alberti G et al in at least the following 8 major points, which distinguish it from that by Alberti G et al and make it a unique contribution to the field.

 (1) We have shown the differential Hsps expression in normal and BRCA tissue, allowing readers to gain a more intuitive understanding of the association between heat shock proteins and breast cancer through the FPKM values presented in the table.

 (2) For Hsp27, we have described in greater detail the mechanism by which Hsp27 and its post-translational modifications (such as the specific phosphorylation sites) affect breast cancer progression and have also listed the signaling regulation involved. In addition, our review not only proposed the inhibition of Hsp27 as a method for breast cancer treatment, but also provided readers with a list of results obtained from the research on inhibiting Hsp27 in breast cancer cells, which lead to the conclusion that “Hsp27 inhibition in breast cancer appears as a promising breast cancer treatment”.

(3) In our review, we have shown the roles of Hsp40 family in the progression of breast cancer, and proposed that “Hsp40 family may have dual functions in breast cancer, both as tumor suppressor and as tumor promoter”.

(4) For Hsp60, we systematically summarized the roles of Hsp60 in breast cancer cells in different subcellular locations, and pointed out that “Hsp60 has a prognostic value and should be inhibited in breast cancer management”; in addition, we summarized the effects of interfering or inhibiting Hsp60 on breast cancer cells.

(5) For Hsp70, in addition to introducing the effects of different subcellular localization and post-translational modification of Hsp70 on breast cancer cells, the notion that “inhibition of Hsp70 could bring benefit to breast cancer patients” was proposed in our review. We also summarized the existing classification of Hsp70 inhibitors (targeting Hsp70 expression, Hsp70ATPase, Hsp70-ATP affinity, HSC70-BAG-1 interactions, and Hsp70-BAG-3 interactions, respectively), and described the effect of these inhibitors on breast cancer cells. Finally, we added that “However, so far none of these inhibitors on the expression or activities of Hsp70 have either been approved for clinical oncology or demonstrated encouraging results in clinical trials due to the fact that they also affect the viability of normal cells.”

(6) For Hsp90, we enumerated the relationship between Hsp90 and breast cancer cells, but focused on showing the advantage and disadvantage of the existing N-terminal domain and C-terminal domain Hsp90 inhibitors as breast cancer therapeutics. Our review concluded that “It should be apparent that Hsp90 is an exciting new therapeutic target, inhibition of which by inhibitors delivers a combinatorial attack on multiple oncogenic pathways and on many hallmark traits of breast cancer”.

(7) In our review, we have shown the association between Hsp110 and breast cancer development, and proposed that “owing to the high chaperoning and immunological activity, recombinant Hsp110 is very promising for breast cancer immunotherapy”.

(8) In the Conclusion, our review concluded that “Among Hsps proteins, Hsp60, Hsp70 and Hsp90 are present on the surface of tumor cells and are released by the cells through exosomes, making them potential prognostic biomarkers that are easily accessible. Similarly, Hsp27, Hsp40 and Hsp110 are also associated with breast cancer development. However, Hsp40 has been reported to have dual functions in breast cancer; in addition, Hsp110 is mainly considered as adjuvants to breast cancer antigens in immunotherapy.” We also pointed out the reason why no Hsps inhibitors have yet been approved by the FDA for clinical use in patients with breast cancer, and put forward the corresponding improvement suggestions.

 To recap, our review has put forward many new points that have not been covered by similar reviews published previously including the one by Alberti G et al. Therefore, we believe our manuscript is worth publication in IJMS.

Comment 2: Comments on the Quality of English Language

English grammar is poor and deserves a major overhaul.

Response: We have gone over the manuscript several times and corrected the grammatic errors we have identified. We also have the manuscript checked by several colleagues who are skillful in English writing. In addition, the manuscript has undergone English language editing by MDPI. All changes made to the manuscript are highlighted in red for easy tracking.

Reviewer 2 Report

Comments and Suggestions for Authors

Article devoted to heat shock proteins and their role in the processes of proliferation, metastasis and apoptosis. The authors focus their attention on six families of these proteins. In my opinion, the manuscript is an interesting and well prepared. After reading it, I would like to point out a few editorial errors and share my comments regarding this study.

Keywords – is Breast cancer; should be breast cancer - the uppercase letter should be replaced with a lowercase one.

Introduction - Table 1 and Figure 1 are a repetition of the same results. It seems to me that the authors should decide on one type of presentation. In my opinion, Table 1 is more readable and I would opt for this form. Genes should be written in italics, while proteins should be written without italics and, as the authors use in the text, they are written with a capital letter.

Paragraph 2. The Biological function of Hsps in Breast Cancer. There are a number of abbreviations used in this paragraph. Some of them are very well explained, e.g. line 103 - ... the epidermal growth factor (EGF)-induced ... , but there are also abbreviations in which their expansion is not given, e.g. MST1 line 98, DCs line 102, UVC line 122. If more abbreviations are used in the text, they should also be explained when they are first quoted in the text. Abbreviations for kinases, e.g. in Figure 2, are self-explanatory.

Page 6 & 8, Figure 4. Line 248. The spelling of the p53 gene and protein is also debatable. In the publications cited by the authors it is written with a lowercase letter, however, taking into account that human genes and proteins should be written with a capital letter and the convention used by the Authors earlier with Hsps, perhaps P53 should be written with a capital letter - protein without italics P53 and the gene with italics P53.

Page 8. I would also explain the abbreviation ER (endoplasmic reticulum probably) on page 8, especially since the abbreviation CFEA is further explained.                                                                                    „ER specific localization of Hsp60 plays a pro-apoptotic role by down-regulating line 223                                      anti-apoptotic protein XIAP expression in chloroform fraction of E. alba (CFEA)-induced – line 224         cell death [53].

Page 8. Line 239 is - breast cancer cells[61]. There is no space between the quoted numer.

Page 10. Line 269 … while gene silencing of Hsp70 … - This is a mental shortcut. Rather, it is the silencing of the gene encoding Hsp70.

Line 300 & 301 Sulphoraphane, a natural isothiocyanate inhibiter on Hsp27 …,  - rather inhibitor of Hsp27 …

Page 12 line 362. … higher NPI prognostic scores - I would explain the acronym NPI and similarly line 369 BCSCs (breat cancer steem cells) as well as  on page 14 line 429 - PP2A.

References. The cited work, numer 50 contains names and the first letter of the surname – should be – IZBICKA E.,  CAMPOS D., CARRIZALES G. and PATNAIK M.

There is a similar error in work no. 70 and 92.

In summary, I believe that the work submitted for review is interesting, it summarizes the current state of knowledge about the described Hsps proteins and, after minor editing corrections, it should be published IJMS.

Comments on the Quality of English Language

Minor editing of English required.

Author Response

Responses to Reviewer #2:

Comment 1: Article devoted to heat shock proteins and their role in the processes of proliferation, metastasis and apoptosis. The authors focus their attention on six families of these proteins. In my opinion, the manuscript is an interesting and well prepared. After reading it, I would like to point out a few editorial errors and share my comments regarding this study.

Response: Thank you very much for your efforts in reviewing our manuscript. Those comments are all valuable and very helpful for revising and improving our paper, as well as the important guiding significance to our researches. We have studied comments carefully and have made correction which we hope meet with approval.

Comment 2: Keywords – is Breast cancer; should be breast cancer - the uppercase letter should be replaced with a lowercase one.

Response: Thanks for your comments. We have made the corrections and highlighted them in the revised manuscript (Line 19 on page 1).

Comment 3: Introduction - Table 1 and Figure 1 are a repetition of the same results. It seems to me that the authors should decide on one type of presentation. In my opinion, Table 1 is more readable and I would opt for this form. Genes should be written in italics, while proteins should be written without italics and, as the authors use in the text, they are written with a capital letter.

Response: Sincerely thanks for your kind advice. We have removed Figure 1 and corrected the font styles for gene and protein names in Table 1 (on page2).

Comment 4: Paragraph 2. The Biological function of Hsps in Breast Cancer. There are a number of abbreviations used in this paragraph. Some of them are very well explained, e.g. line 103 - ... the epidermal growth factor (EGF)-induced ... , but there are also abbreviations in which their expansion is not given, e.g. MST1 line 98, DCs line 102, UVC line 122. If more abbreviations are used in the text, they should also be explained when they are first quoted in the text. Abbreviations for kinases, e.g. in Figure 2, are self-explanatory.

Response: Following the reviewer’s suggestion, we spelled out the full names for the abbreviations when they are first quoted in the text, and highlighted the changes in the revised manuscript (Line 62 on page 2; line 104, line 108-109, line 111-112 on page 3; line 129 on page 4; line 224 on page 7; line 391-392 on page 12; line 466 on page 14).

Comment 5: Page 6 & 8, Figure 4. Line 248. The spelling of the p53 gene and protein is also debatable. In the publications cited by the authors it is written with a lowercase letter, however, taking into account that human genes and proteins should be written with a capital letter and the convention used by the Authors earlier with Hsps, perhaps P53 should be written with a capital letter - protein without italics P53 and the gene with italics P53.

Response: Thanks for your comments. We have made the corrections following the reviewer’s suggestion, and highlighted them in the revised manuscript (Line 194, line 195, line 201 on page 6; line 236, line 252 on page 8; figure 4 on page 7; figure 6 on page 8).

Comment 6: Page 8. I would also explain the abbreviation ER (endoplasmic reticulum probably) on page 8, especially since the abbreviation CFEA is further explained.                                                                                    „ER specific localization of Hsp60 plays a pro-apoptotic role by down-regulating line 223                                      anti-apoptotic protein XIAP expression in chloroform fraction of E. alba (CFEA)-induced – line 224   cell death [53].

Response: Thanks for your comments. We have explained the abbreviation ER in the text, and highlighted them in the revised manuscript (Line 224 on page 7).

Comment 7: Page 8. Line 239 is - breast cancer cells[61]. There is no space between the quoted numer.

Response: Thanks for your comments. We have made the corrections (Lines 260 on page 8) in the revised manuscript.

Comment 8: Page 10. Line 269 … while gene silencing of Hsp70 … - This is a mental shortcut. Rather, it is the silencing of the gene encoding Hsp70.

Response: Thanks for your comments. We followed the comments and made the revisions (Lines 290 on page 9).

Comment 9: Line 300 & 301 Sulphoraphane, a natural isothiocyanate inhibiter on Hsp27 …,  - rather inhibitor of Hsp27 …

Response: Thanks for your comments. We have made the corrections (Lines 322-323 on page 10) in the revised manuscript.

Comment 10: Page 12 line 362. … higher NPI prognostic scores - I would explain the acronym NPI and similarly line 369 BCSCs (breat cancer steem cells) as well as on page 14 line 429 - PP2A.

Response: We have explained the abbreviation NPI, PP2A, and BCSCs in the revised manuscript (Line 391-392 on page 12; line 466 on page 14; and line 105 on page 3, respectively).

Comment 11: References. The cited work, numer 50 contains names and the first letter of the surname – should be – IZBICKA E.,  CAMPOS D., CARRIZALES G. and PATNAIK M.

There is a similar error in work no. 70 and 92.

Response: We thank the reviewer for pointing out the errors. We have corrected this (no. 50; no. 69 and no. 91 in the revised version) and all other similar errors and highlighted them in the references.

Comment 12: Comments on the Quality of English Language

Minor editing of English required.

Response: We thank the reviewer for the kind suggestion, and have gone over the manuscript carefully to correct typos and grammatical errors. In addition, the manuscript has undergone English language editing by MDPI. All changes made to the manuscript are highlighted in red for easy tracking.

Reviewer 3 Report

Comments and Suggestions for Authors

The authors present a review manuscript which deals with heat shock proteins in Breast Cancer. This is an interesting area of research. There are a few reviews in this area which deal with heat shock proteins in cancer. 

The abstract is fairly well writen and very concise. Overall it sums up the aims/objectives of the current review quite well. 

Introduction - generally the first part of the introduction is well written and referenced. I'm not sure of the accessibility of Table 1 and Figure 1 though - these are quie difficult to interpret and there is no sufficient information to indicate the significance of the Table/Figure. Could there be additional explanation, highlighting/annotation to make it more meaningful?

Section 2. I think if possible throughout the paper there could be smaller focused paragraphs

Not sure what Figure 4a is really showing, possibly requires expansion of description/annotation

There are quite a few acronyms used that are not defined throughout the text, or only partially defined. 

Figures are often split into multiple sub figures. Maybe it would improve the clarity of the description in the text and legends to have them separately where eacxh part is not directly linked to the other?

Not sure that Figure 6A is imparting much information regarding the structure etc. Same can be said for some of the preceding figures. The diagrams should really aid understanding beyond the text description.

In my opinion Figure 7 is quite confusing and needs some work to clarify. For example the series of arrows in part A. The arrows in part C also are not that clear.

Conclusions section  - weaken? rather than weak "The 463 efficacy of current Hsps inhibitors is relatively low, and inhibition of one Hsps member 464 could upregulate other Hsps via negative feedback to weak the overall therapeutic effect 465 of the inhibitors."

In my opinion the review may benefit from a brief summary of future directions for establishing these proteins as important markers of breast cancer and potential targets for therapy etc.

Comments on the Quality of English Language

English generally is fine. The main issue is structural/organisation from my perspective.

Author Response

Responses to Reviewer #3:

Comment 1: The authors present a review manuscript which deals with heat shock proteins in Breast Cancer. This is an interesting area of research. There are a few reviews in this area which deal with heat shock proteins in cancer.

Response: Thank you very much for your affirmation.

Comment 2: The abstract is fairly well writen and very concise. Overall it sums up the aims/objectives of the current review quite well.

Response: Thanks for your positive comments.

Comment 3: Introduction - generally the first part of the introduction is well written and referenced. I'm not sure of the accessibility of Table 1 and Figure 1 though - these are quie difficult to interpret and there is no sufficient information to indicate the significance of the Table/Figure. Could there be additional explanation, highlighting/annotation to make it more meaningful?

Response: We thank the reviewer for the suggestion. We have added some additional explanation for Table 1 in the revised manuscript (Line 58, line 63-66 on page 2).

Comment 4: Section 2. I think if possible throughout the paper there could be smaller focused paragraphs.

Response: We thank the reviewer for the suggestion. In the Section 2, our description of each heat shock proteins (Hsps) is roughly divided into three parts. First, we described the structure of each member of Hsps, and their post-translational modifications; secondly, we described the association between Hsps and breast cancer development (the order of description takes into account the subcellular localization where Hsps exerts their function in breast cancer regulation, the post-translational modification of Hsps in regulating breast cancer development, the order of publication of references, etc.); and finally, we proposed the role of the heat shock protein in breast cancer and described the application of existing interference or inhibitors of Hsps.

Throughout the manuscript, we have kept our description as concise as possible. On the other hand, we also have to furnish enough details so that the readers could get a comprehensive and coherent understanding of the points being discussed. Hopefully the reviewer could understand and agree with us.

Comment 5: Not sure what Figure 4a is really showing, possibly requires expansion of description/annotation.

Response: Thanks for your comments. We have added the description in the figure legend for Figure 4a (Figure 5 in the revised version) in the revised manuscript (Line 231-233 on page7).

Comment 6: There are quite a few acronyms used that are not defined throughout the text, or only partially defined.

Response: Thanks for your comments. Following the reviewer’s suggestion, we have explained the abbreviations when they are first quoted in the text, and highlighted them in the revised manuscript (Line 62 on page 2; line 104, line 108-109, line 111-112 on page 3; line 129 on page 4; line 224 on page 7; line 391-392 on page 12; line 466 on page 14).

Comment 7: Figures are often split into multiple sub figures. Maybe it would improve the clarity of the description in the text and legends to have them separately where each part is not directly linked to the other?

Response: We thank the reviewer for the constructive suggestion, and have split the figures into sub figures. The figure legends were also modified accordingly in the revised manuscript.

Comment 8: Not sure that Figure 6A is imparting much information regarding the structure etc. Same can be said for some of the preceding figures. The diagrams should really aid understanding beyond the text description.

Response: Following the reviewer’s suggestion, we have added some details in the legends of Figure 6A and also modified the figure to aid understanding (Figure 9 in the revised version, Line 381-384 on page 12). In addition, we have redrawn Figure 5A and revised the corresponding legends (Figure 7 in the revised version, Line 285-287 on page 9).

Comment 9: In my opinion Figure 7 is quite confusing and needs some work to clarify. For example the series of arrows in part A. The arrows in part C also are not that clear.

Response: Thanks for your comments. We have added the interpretation of the arrows in the figure legend for Figure 7A (Figure 11 in the revised version) in the revised manuscript (Line 459 on page 14). In addition, we have added some details in the legends of Figure 7C (Figure 12 in the revised version) in the revised manuscript (Line 477-483 on page 15).

Comment 10: Conclusions section  - weaken? rather than weak "The 463 efficacy of current Hsps inhibitors is relatively low, and inhibition of one Hsps member 464 could upregulate other Hsps via negative feedback to weak the overall therapeutic effect 465 of the inhibitors."

Response: We thank the reviewer for the comments, and have made the corrections (Line 503 on page 15) in the revised manuscript.

Comment 11: Comments on the Quality of English Language

English generally is fine. The main issue is structural/organisation from my perspective.

Response: We would like to thank the reviewer for the positive comments and valuable suggestions. We have revised the manuscript carefully based on these comments and believe that the revised version has improved much in quality.

Round 2

Reviewer 1 Report

Comments and Suggestions for Authors

The present version of the manuscript has been ameliorated in terms of written English. However, authors demonstrated little consideration of the comments provided during the first review of the manuscript, since no implementation has been made in terms of content organisation and critical discussion.

Author Response

Response: Thanks for your comments. The overall content organisation of the manuscript is referred to the layout of some Reviews in IJMS[1-8] (the main body part: mainly summarize the findings of the original research cited; the last part: conclusion and discussion), we used relatively few critical languages in the Section 2 in order to truthfully overview the findings of the original research cited. Most of the critical language was found in Section 3(Conclusions), where we not only systematically summarized the relationship between heat shock protein and breast cancer, but also pointed out the problems existing in the prevention and treatment of breast cancer through targeting Hsps, and finally we put forward a prospect for the field.

  1. Al Salhi, Y.; Sequi, M. B.; Valenzi, F. M.; Fuschi, A.; Martoccia, A.; Suraci, P. P.; Carbone, A.; Tema, G.; Lombardo, R.; Cicione, A.; Pastore, A. L.; De Nunzio, C., Cancer Stem Cells and Prostate Cancer: A Narrative Review. Int J Mol Sci 2023, 24, (9):7746.
  2. Alalawi, S.; Albalawi, F.; Ramji, D. P., The Role of Punicalagin and Its Metabolites in Atherosclerosis and Risk Factors Associated with the Disease. Int J Mol Sci 2023, 24, (10):8476.
  3. El Hage, R.; Al-Arawe, N.; Hinterseher, I., The Role of the Gut Microbiome and Trimethylamine Oxide in Atherosclerosis and Age-Related Disease. Int J Mol Sci 2023, 24, (3):2399.
  4. Zhao, W.; Wang, L.; Wang, Y.; Yuan, H.; Zhao, M.; Lian, H.; Ma, S.; Xu, K.; Li, Z.; Yu, G., Injured Endothelial Cell: A Risk Factor for Pulmonary Fibrosis. Int J Mol Sci 2023, 24, (10):8749.
  5. Baggio, C.; Bindoli, S.; Guidea, I.; Doria, A.; Oliviero, F.; Sfriso, P., IL-18 in Autoinflammatory Diseases: Focus on Adult Onset Still Disease and Macrophages Activation Syndrome. Int J Mol Sci 2023, 24, (13):11125.
  6. Szrok-Jurga, S.; Turyn, J.; Hebanowska, A.; Swierczynski, J.; Czumaj, A.; Sledzinski, T.; Stelmanska, E., The Role of Acyl-CoA beta-Oxidation in Brain Metabolism and Neurodegenerative Diseases. Int J Mol Sci 2023, 24, (18):13977.
  7. Mondelli, M. U.; Ottolini, S.; Oliviero, B.; Mantovani, S.; Cerino, A.; Mele, D.; Varchetta, S., Hepatitis C Virus and the Host: A Mutual Endurance Leaving Indelible Scars in the Host’s Immunity. International Journal of Molecular Sciences 2023, 25, (1):268.
  8. Rusek, M.; Smith, J.; El-Khatib, K.; Aikins, K.; Czuczwar, S. J.; Pluta, R., The Role of the JAK/STAT Signaling Pathway in the Pathogenesis of Alzheimer's Disease: New Potential Treatment Target. Int J Mol Sci 2023, 24, (1): 864.

Reviewer 3 Report

Comments and Suggestions for Authors

The authors made changes to the manuscript which in my opinion improve the clarity. From my perspecive it is an interesting review and should be published. 

Author Response

Response: Thank you very much for your efforts in reviewing our manuscript. Thanks for your affirmation.